# Effects of Nitrogen Deficiency and Resupply on the Absorption of Mineral Nutrients by Tangor Cultivar ‘Shiranuhi’ (*Citrus unshiu* × *C. sinensis*) Grown in a Hydroponic System

**DOI:** 10.3390/plants11182351

**Published:** 2022-09-08

**Authors:** Seong Heo, Won-Pyo Park

**Affiliations:** 1Department of Horticulture, Kongju National University, Yesan 32439, Korea; 2Faculty of Bioscience and Industry, College of Applied Life Science, Jeju National University, Jeju 63243, Korea

**Keywords:** *Citrus unshiu* × *C. sinensis*, hydroponics, mineral nutrient, nitrogen deficiency, tangor

## Abstract

Tangor (*Citrus unshiu* × *C. sinensis*) cultivars obtained through interspecific hybridization have gained popularity in various countries, including South Korea. However, tangor has a relatively short history of cultivation and thus has been less well researched in terms of physiology and opping technology compared to mandarin. In addition, although tangor grows more vigorously than mandarin and thus has high nutrient demands and longer harvest time, it is more prone to various physiological disorders. Furthermore, the demand for nutrients could vary depending on the cultivars even within the same species. Thus, the current study examined the effects of nitrogen deficiency and resupply on the uptake of mineral nutrients using a specific cultivar ‘Shiranuhi’ as a case study. In this study, the tangor cultivar ‘Shiranuhi’ was cultivated in a hydroponics system, which allows the control of nutrient contents, and changes in mineral nutrient contents under nitrogen deficiency and nitrogen resupply were examined. Based on this, the results show the straightforward relationship between nitrogen and other mineral nutrients under a hydroponic system applying the cultivation of tangor. This implies that the hydroponics system can be applied to tangor cultivation and efficiently and widely expanded in Northeast Asia, and the income of growers will increase through the production of high-quality fruits.

## 1. Introduction

Nitrogen (N) is regarded as the most essential mineral nutrient that influences plant growth and crop yield and quality [1]. It is a major constituent of chlorophyll, nucleic acids, amino acids, proteins, plant hormones, and coenzymes, and it is important for the synthesis of primary and secondary metabolites such as phenolic and aromatic compounds [2,3]. In addition, N affects cell division; the growth of vegetative and reproductive organs, including buds, flowers, leaves, and stem; tree vigor; the absorption and distribution of other nutrients during flowering and fruit set [1]; and even fruit bearing in the following year (biennial bearing). In particular, early growth of fruits and leaves is highly influenced by N, and this nutrient is required for the consequent leaf (photosynthetic) area and fruit expansion. Therefore, N fertilizers are widely supplied to increase fruit yield and enhance fruit quality. The effects of N deficiency in citrus have been mainly investigated in sweet orange and mandarin. N deficiency in citrus causes leaves to turn pale green or yellowish and, eventually, shed [4]. The leaf veins turn lighter in color, and tree growth and development are restricted. The fruits harden, with lower acidity and poorer storability. Finally, fruit yield and quality decrease [5,6,7,8,9].

Since the beginning of citrus fruit cultivation, substantial effort has been put into promoting reasonable fertilization management to increase fruit quantity and quality. Early studies focused mainly on the effects of N deficiency on fruit quality and quantity. Alva and Paramasivam [10] and Tachibana and Yahata [11] showed that, while fruit quantity increased with an increasing amount of fertilizer, fruit quality remained unchanged. The percentage of dehiscent fruits and fruits with poor coloration increased with an increasing amount of fertilizer. The sugar content was unaffected, while the acid content increased. More recent studies have reported on the effects of N deficiency or uptake [3,12]. In *C. sinensis* and *C. grandis*, rapid soil or foliar N application is necessary for recovery from N deficiency symptoms [3], and N application is required in *C. unshiu* and *C. sinensis* [12]. For tangor (*C. unshiu* × *C. sinensis*), which has different physiological characteristics from mandarin and sweet orange, cultivation techniques have not yet been established and there are not many studies on mineral uptake in tangor. As tangor grows more vigorously than mandarin and sweet orange, its nutrient demands are greater than those of mandarin, and the harvest time is longer than that of *C. unshiu*, which suggests that tangor fruits may be more prone to physiological disorders [5,13,14]. Notably, the fruit peel puffing disorder has caused problems in Jeju-do, South Korea, and there is an urgent need for physiological studies to clarify whether mineral nutrient deficiency is at the root of this disorder.

As it is generally difficult to control and measure the absorption of mineral nutrients under soil cultivation, hydroponics systems are widely used [15,16,17,18,19,20,21,22,23,24]. These systems allow easy nutrient solution management and optimal supply of mineral elements in soluble form [20,21,22,23,24,25]. Therefore, hydroponics is suitable for determining the proper concentrations of nutrients required by plants and investigating the effect of nutrient deficiencies. In this study, the tangor cultivar ‘Shiranuhi’ was cultivated in a hydroponics system under various levels of N supply, and changes in the contents of other mineral nutrients after N deficiency and N resupply were examined with the aim to identify nutrients that have a synergetic or antagonistic relationship with N in tangor and define the characteristic mineral nutrient contents under N deficiency and N resupply.

## 2. Results and Discussion

### 2.1. Symptoms of N Deficiency in Leaves of ‘Shiranuhi’

Figure 1A shows a comparison of leaves of plants from the control and N deficiency (ND) groups at various time points during N deficiency treatment. The earliest symptom of N deficiency, whereby the leaves turned light green in color, was visible 40 days after N deficiency (40 DAND). At 50 DAND, the veins and midrib turned yellow, and at 100 DAND, the leaf lamina showed chlorosis. At 120 DAND, the entire leaf showed chlorosis, which is a typical symptom of N deficiency [26], and the N content was further reduced to 1.21%. When foliar applications of 0.5% urea were given as of 54 DAND in the foliar application (FA) group (after the first symptoms of N deficiency appeared), or when the complete nutrient solution was resupplied to the roots in solution resupply (SR) group, the chlorosis disappeared, and the leaf color returned from light to dark green between 80 and 100 DAND (data not shown). In the ND group, at 120 DAND, N deficiency was apparent from the stunted growth of the aboveground as well as belowground plant parts and the N deficiency symptoms in the leaves, including chlorosis around the veins, eventually induced abscission (Figure 1A).

### 2.2. Effects of N Deficiency and Resupply on Nitrogen and Chlorophyll Contents of Leaves

The N content and chlorophyll content (based on SPAD value) were measured as the leaves turned yellow in the ND group (Figure 1B). The continuous N supply in the control group completely prevented the deficiency symptoms as shown in Figure 1A, and the N content was adequately maintained at around 3.1% (Figure 1C). However, the N content in the ND group continuously decreased to reach 1.1% at 120 DAND (*p* = 5.82 × 10^−3^, Figure 1C). The SR group, with N resupply at 54 DAND, showed a steep increase in N content at 60 DAND (*p* = 3.87 × 10^−4^), and the N content was restored to the level of the control group by 120 DAND. Similarly, the FA group showed a gradual increase in N content by 60 DAND, although the increase was not as steep as that in the SR group and did not reach the control level (Figure 1C).

SPAD values were determined as an indicator of photosynthetic efficiency in the leaves during the experimental period (Figure 1B). At 40 DAND, SPAD values in the control group were adequately maintained at ≥85.2 (*p* = 0.269), whereas those in the ND group steadily decreased to 17.0 by 120 DAND (*p* = 4.43 × 10^−5^). This trend was in line with the pattern in N content (Figure 1C). Indeed, a high correlation was found between the SPAD values and N content in the ND group (r^2^ = 0.646). Analysis of the correlation between the SPAD values and N contents in both the ND and control groups revealed that the coefficient of determination was slightly lower (r^2^ = 0.593) (Figure 2).

Although the SPAD values also tended to decrease due to N deficiency during 54 DAND, the decrease was not as large as that in the N content, and the increase in N content after N resupply was not reliably reflected in the SPAD values. At 120 DAND, the N content showed significant differences among the control, SR, and FA groups, whereas the SPAD values did not.

At 54 DAND, the chlorophyll in the leaf is rapidly degraded and the components are transferred to other plant parts for reuse, but with the N resupply at 54 DAND, the chlorophyll content was rapidly restored. Thus, regarding the nitrogen use efficiency in citrus fruits, N seems to be preferably allocated to chlorophyll synthesis. Further, as N supply through roots induces effects as fast as foliar application, the former is expected to allow more effective recovery from physiological disorders caused by N deficiency.

### 2.3. Effects of N Deficiency and Resupply on the Absorption of Other Mineral Nutrients from Leaves

Next, we determined the effects of N deficiency on the absorption of other mineral nutrients from the leaves. The results are shown in Figure 3. In the ND group, the P content showed a significant increase and was consistently higher than that in the control group at all time points measured, except 20 DAND. This finding indicates that P absorption is significantly increased by N deficiency. When we compared the SR and FA groups, we found that plants in the SR group showed a high level of P absorption until 40 DAND. This trend was maintained even after N resupply, and the SR group showed a higher P content than any other group at 60 DAND. Thus, P absorption is presumed to have increased under the influence of N deficiency, but after N resupply (54 DAND), the P content decreased to a level close to the control level.

Seasonal temperature changes during the cultivation period are presumed to have had an effect. As shown in Figure 4, the temperatures inside the greenhouse, pot, nutrient solution, and soil were the highest between late August and early September (60 DAND). The temperature steadily declined thereafter to reach a low around late October, followed by a slight increase in early November (120 DAND). This pattern was notably similar to that of the nutrient contents in the control and SR groups. The sudden decrease in P content in the SR group to a level similar to that in the control group at 80 DAND is presumed to be due to the restoration of normal N metabolism after N resupply.

In the FA group, P contents continuously increased after 54 DAND to reach 0.39% by 120 DAND and were substantially higher than those in the other groups. The P content pattern was similar to that in the ND group rather than that in the control group, which is likely because, despite the rapid N supply through foliar application, the supply was localized to the leaves and the actual N content was lower than that in the control group (Figure 1C).

K absorption increased after N supply (Figure 3B). Compared to the other groups, the ND group showed the lowest K contents at all time points measured, with significant differences. In the control group, the K content was the highest at 60 DAND (9/10), reached the lowest level at 80 DAND (9/30), and steadily increased until 120 DAND. This pattern was similar to that of the P content in the control and SR groups, and is also presumed to have been influenced by seasonal temperature changes. The SR group showed a similar K content to the control group until 54 DAND (*p* > 0.05). However, after N resupply, the K content showed a sudden increase at 60 DAND, indicating that absorption of N and K was simultaneously promoted as the nutrients were resupplied.

The Ca content and absorption were reduced by N deficiency (Figure 3C). The ND group showed a significantly lower Ca content than the control group due to insufficient N supply. In the SR group, the Ca content strongly increased after N resupply to a level higher than that in the control. In the FA group, in contrast, the Ca content slightly increased at 40 DAND and then remained stable until 120 DAND. The Ca level in this group was similar to that in the ND group, but significantly different from that in the control group. Thus, Ca absorption was not increased by N resupply through foliar application. Only in the condition of sufficient N supply to the roots, Ca absorption was increased, and unlike the K content, the Ca content did not increase through foliar application for N supply.

The absorption of Mg was enhanced by N deficiency and reduced by N supply (Figure 3D). In the control group, Mg absorption tended to decrease with N supply. At 20 DAND, the Mg content was substantially higher than that in the other groups, but at 120 DAND, the content was lower than that in the ND group, which appeared to show an opposite pattern to the control group. Although N deficiency did not have a significant effect on the Mg content, N supply seemed to interact antagonistically with Mg absorption.

As shown in Figure 3E, the B content at 20 DAND differed significantly between the control and ND groups, whereas it decreased to similar levels at 60 DAND. Thereafter, the B content increased in both groups, with no significant difference. The B content levels in the SR and FA groups were substantially higher than those in the control and ND groups. This is presumably due to increased absorption of B with N resupply following N deficiency. Thus, B absorption increased only with N resupply after deficiency, implying conditional synergism.

The Zn content was not influenced by the N content (Figure 3F). The highest Zn content of 51.6 mg kg^−1^ was observed in the control group at 20 DAND, but the Zn content steadily decreased thereafter. In the ND group, the Zn content also decreased to <20 mg kg^−1^ at 40 DAND and remained stable thereafter. In the FA group, the Zn content remained <20 mg kg^−1^, irrespective of N resupply. Thus, the Zn demand in tangor is high before 20 DAND and decreases thereafter.

The Mn content was low in all groups at 20 DAND, but showed a sudden increase at 40 DAND. This is presumed to be due to the increased demand for Mn under the influence of temperature changes. Hence, Mn absorption shows synergism with N supply, whereby the former increases under the influence of the latter.

As for the Fe content, like for the Zn content, the highest content and the largest variation were observed at 20 DAND. Thereafter, the Fe content remained around 40 mg kg^−1^ in all groups but the control group at 60 DAND. The SR and FA groups, irrespective of N resupply, showed no significant difference from the ND group at all time points except 120 DAND.

Upon N deficiency, the Cu content was lower, albeit not significantly, than that in the control group. After 54 DAND, the Cu content did not vary significantly in all groups at all time points except 80 DAND.

### 2.4. Pearson Correlation Analysis

The correlation between N and other mineral nutrients was analyzed using Pearson correlation analysis (Figure 5). High correlations were found between N and K (SR: r = 0.826, control: r = 0.681), N and Ca (SR: r = 0.825), and N and B (SR: r = 0.631). As shown in Figure 3, the SR and control groups showed a typical pattern of temperature-dependent nutrient absorption, with nutrient contents being high at 60 DAND (9/10), decreasing at 80 DAND (9/30), and increasing thereafter. This trend similarity resulted in the above high correlations. In contrast, a negative correlation was found between N and Zn in the control group (r = −0.613), and a high positive correlation of 0.857 was found between P and K in the FA group. The N supply through foliar application increases the contents of P and K, which are mainly used in chlorophyll synthesis, which is presumed to account for the high SPAD value. A negative correlation was found between P and Ca in the control group (r = −0.641) because, while N supply reduced the absorption of P, it increased the absorption of Ca.

The correlation between P and Mg was high in all groups except the control group. As with P, the content of Mg was increased by N deficiency and decreased by N supply. The contents of P and Mg were higher in the SR and FA groups with N deficiency until 54 DAND and in the ND group without N supply than in the control group with continuous N supply. Thus, high P and Mg levels may be important markers of N deficiency and can be useful for identifying N deficiency in plants. In contrast to the SR and control groups, the FA group maintained high P and Mg contents, which likely increased the photosynthetic efficiency.

High correlations were found between K and Mg, K and Zn, and K and Fe in the FA group. As the correlations between P and K and between P and Mg were high in the FA group, a strong positive correlation was also found between K and Mg. However, a strong negative correlation was found between K and other metallic elements, such as Zn and Fe, because the N resupply reduced the absorption of Zn and Fe.

Both Ca and Mn showed reduced absorption upon N deficiency. As a result, high correlations were found between Ca and Mn in the ND and FA groups. Given the strong negative correlations of Zn and Fe with K, the correlation between Zn and Fe in the FA group was substantially high. In addition, the ND group showed high correlations for Zn-Fe and Zn-Cu, and the control group showed a high correlation for Zn-Cu. The metallic elements were found not to be significantly influenced by N content, but to show an inverse relationship to the temperature change pattern shown in Figure 4. As for the correlations between Mn and Fe and Mn and Cu in the FA group, Mn absorption increased with N supply, whereas Fe and Cu tended to decrease. Thus, N shows synergism with Mn, but antagonism with Fe and Cu.

### 2.5. Principal Component Analysis

In the control group, all nutrients, including N, were continuously supplied via the nutrient solution, whereas in the ND group, all nutrients excluding N were continuously supplied for the experimental period of 120 days. Hence, as shown in the biplot in Figure 6, the 95% confidence ellipse for the control group clearly differed from that for the ND group. The control group showed a smaller confidence ellipse than the other groups due to smaller variation. As shown in Figure 3, in the control group, the contents of K, Ca, Mn, and B showed synergetic relationships with the N content. Accordingly, plans in the control group were characterized by the high contents of N, K, Ca, Mn, and B. In contrast, in the ND group, the strongest contributions to PC1 were from P (0.822), Mg (0.813), and K (0.767) and those to PC2 from Zn (−0.832), Fe (−0.761), and N (−0.711). In brief, the ND group was characterized by high P and Mg contents and low Zn and Fe contents due to N deficiency.

The SR group, which was supplied all nutrients including N via the nutrient solution to the roots at 54 DAND, showed the largest confidence ellipse and, thus, the largest variation. PC1 in the SR group was affected by the same nutrients as in the control, whereas PC2 was mostly affected by P and Mg, as in the ND group. The confidence ellipse of the SR group overlapped with those of all other groups, and particularly, included that of the control group.

The confidence ellipse of the FA group largely overlapped with that of the ND group. Despite the N resupply, the FA group was significantly affected by the contents of P and Mg and showed characteristic low contents of metallic elements, such as Fe and Zn. The pattern was highly similar to that observed for the ND group. Although the influence of N was not as significant as that in the control and SR groups, in the FA group, PC1 was significantly affected by Ca and Mn, in contrast to the ND group. Thus, as N supply via foliar application induced increases in leaf Ca and Mn contents, physiological disorders due to Ca or Mn deficiency could be prevented. In summary, the SR and FA groups, which experienced N deficiency, exhibited characteristically high contents of P and Mg, despite N resupply.

Tangor is a high-quality fruit that offers both orange and mandarin flavors, forming a high-priced market in South Korea. Currently, citrus growers are choosing a strategy to produce high-quality fruits even in small quantities rather than mass producing low-quality fruits. Due to this trend, the introduction of the hydroponics system has been accelerated in Jeju Island, the main production area for tangor cultivation. In this study, the effects of nitrogen deficiency and resupply, which are directly related to the quality of tangor, on the absorption of other mineral nutrients were confirmed by applying the hydroponics system. In tangor ‘Shiranuhi’, it was found that visible symptoms appeared on the leaves after 40 days of nitrogen deficiency. In addition, if nitrogen was resupplied within 54 days of nitrogen deficiency, it was restored to normal conditions. There was a high correlation between leaf nitrogen content and SPAD value in nitrogen-deficient treatment. This is a theoretical background that can easily predict the nitrogen content by measuring the SPAD value of ‘Shiranuhi’ leaves, and based on this, a gradient boosting model was developed that could most accurately predict nitrogen content among several machine learning models [27]. Although the foliar fertilization (FA group) was expected to show rapid nitrogen recovery in nitrogen-deficient tangor leaves, nitrogen content was rather lower than that in treatment that resupplied nutrient solution directly to the roots (SR group). Therefore, it is necessary to promote to tangor growers that the supply of nutrient solution is more effective in hydroponics systems than foliar fertilization, unlike general cultivation practices. In addition, as shown in Figure 1, FA is not an alternative to SR in terms of long-term nitrogen nutrient management. The correlation of absorption of other mineral nutrients was different according to nitrogen deficiency and resupply, but there was a high correlation between them. This means that it is possible to indirectly determine whether a tangor leaf is deficient in nitrogen by measuring the concentration of other mineral nutrients. Our research group has developed several machine learning models that can identify whether or not a tangor leaf is lacking in nitrogen nutrient [28].

## 3. Materials and Methods

### 3.1. Plant Materials

Three-year-old ‘Shiranuhi’ plants were transferred to a hydroponics system installed inside a greenhouse. The plants were acclimated to the hydroponics system by allowing normal growth in the system for approximately six months. To prevent leaf necrosis caused by Ca deficiency during these six months of cultivation, 0.2% calcium nitrate [5(Ca(NO_3_)_2_·H_2_O)·NH_4_NO_3_] was provided through foliar application at 3-day intervals.

### 3.2. Hydroponics System for Citrus Cultivation

The hydroponics system used in this study was based on the deep flow technique. The system consisted of a pot, a nutrient solution supply equipment, and a nutrient solution supply timer, as shown in Figure 7A.

#### 3.2.1. Pots

The pot used for hydroponic cultivation was a 15 L cylindrical polyethylene container. The pot was placed in the center of a Styrofoam box (60 × 60 × 35 cm) with 5-cm-thick walls, and the surrounding area was filled with soil to support the pot of the hydroponics system (Figure 7B). The bottom of the container was connected to the nutrient solution supply equipment by supply and drainage tubes. A tube to prevent overflow was installed at the top of the pot and connected to the nutrient solution bottle (Figure 7). To monitor temperature changes in the nutrient solution and rhizosphere, temperature sensors were installed near the tree (to measure the temperature inside the greenhouse), in the cultivation container (to measure the rhizosphere temperature), in the nutrient solution bottle, and at 20 cm deep in the soil, and connected to a data logger (Figure 4).

#### 3.2.2. Nutrient Solution Supply

To maintain natural drainage of the nutrient solution and a stable nutrient solution temperature, a Styrofoam box (35 × 45 × 50 cm) was installed at 50 cm below the ground surface and a 20 L nutrient solution bottle was placed in the box (Figure 7A). A hydroelectric pump was installed inside the nutrient solution bottle to supply the solution to the pot (Figure 7A).

#### 3.2.3. Nutrient Solution Supply Timer

The nutrient solution supply time was controlled through a circuit consisting of a 24-h timer and an IC minute timer. Approximately 7 L of nutrient solution was automatically supplied in 2 min 30 s every 40 min during the day and every 70 min during the night (Figure 7).

#### 3.2.4. Nutrient Solution Composition

The nutrient solution compositions are shown in Table 1. To induce N deficiency, the N sources KNO_3_, 5(Ca(NO_3_)_2_·H_2_O)·NH_4_NO_3_, and NH_4_NO_3_ were excluded from the nutrient solution, whereas KCl and CaCl_2_·2H_2_O were added to adjust the K and Ca concentrations (ND solution) (Table 2).

The nutrient solution was adjusted to pH about 6.0 with 1 N NaOH and HCl. During the cultivation, the pH of nutrient solution was periodically checked and adjusted with 1 N NaOH to maintain about 6.0.

### 3.3. Experimental Treatments

From 12 July (0 DAND), twenty days after the first leaf had developed, N supply was restricted for all groups except the control group. In the control group, plants received the complete nutrient solution for 120 days. In the ND treatment group, plants received the ND solution. For the trees that were subjected to restricted N supply until 2 September (54 DAND) and showed symptoms of N deficiency, N was resupplied through the nutrient solution in the SR group or through foliar application in the FA group. In the FA group, plants received foliar applications of 0.5% urea every two days between 54 DAND and 120 DAND. In the SR group, plants received the same nutrient solution as the control group during this period (Figure 8). Five trees were used per treatment, but three trees were selected per treatment depending on tree survival.

### 3.4. Measurement of Chlorophyll Content

The leaf chlorophyll content was measured five repetitions, using a SPAD-502 chlorophyll meter (Konica Minolta Sensing, Osaka, Japan).

### 3.5. Leaf Nutrient Content Analysis

Leaves were collected every 20 days during the N deficiency treatment and dried in a dehydrator at 70 °C for 24 h. The dried leaf samples were placed in a Kjeldahl flask for H_2_SO_4_-H_2_O_2_ degradation. Part of the degraded materials was subjected to total N determination using the Kjeldahl method [29]. The remaining part was filtered and diluted to measure the contents of P, K, Ca, Mg, B, Mn, Zn, Fe, and Cu using inductively coupled plasma optical emission spectrometry (ICP-OES; JY 138 Ultrace, Jobin Yvon, Longjumeau, France).

### 3.6. Statistical Analysis

Data analysis was performed using R software (v.4.1.2, R Foundation for Statistical Computing, Vienna, Austria). Data were analyzed by one-way ANOVA followed by Scheffé’s post-hoc tests using the ‘agricolae’ and ‘doBy’ packages. Pearson correlation coefficients were calculated using the ‘GGally’ package and principal component analysis (PCA) was conducted using the ‘FactoMineR’ and ‘factoextra’ packages to construct biplots. Other graphs were generated using SigmaPlot v.12.0 (Systat Software Inc., Erkrath, Germany).

## 4. Conclusions

The current study utilized one of the tangor cultivars ‘Shiranuhi’, which has not been researched well compared to mandarin in terms of nutrients demands and cultivation methods. In order to examine them, a hydroponics system was used to control the supply of mineral nutrients targeting N deficiency and N resupply after deficiency as a case study before other nutrients are studied. As a result, clear relationships were found between nitrogen and other minerals. This could imply that this specific cultivar could be well managed in the hydroponic system for better product quality. Thus, it would be worth investigating it further for other nutrients using this system to develop a better cultivation system for tangor cultivars.

## Figures and Tables

**Figure 1 plants-11-02351-f001:**
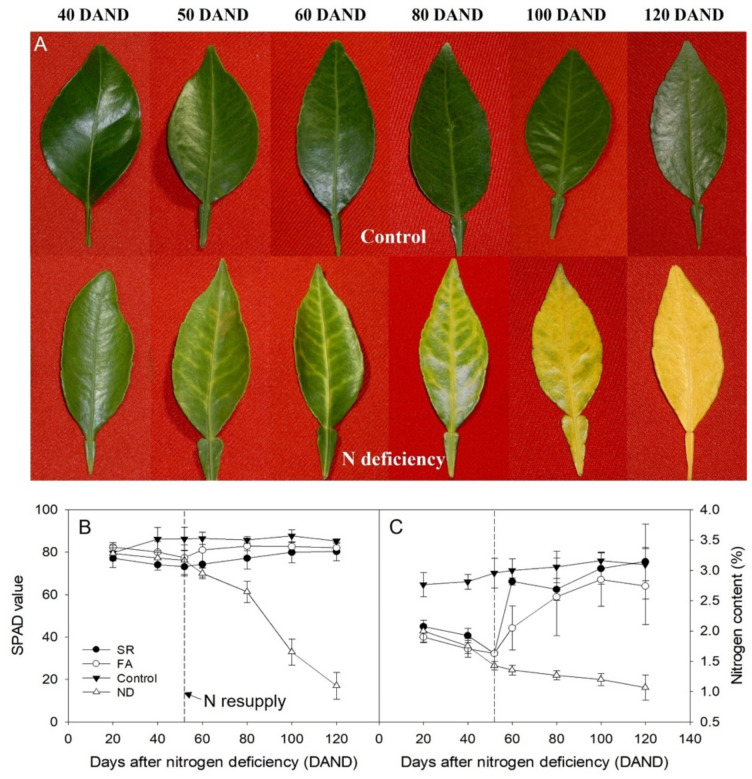
(**A**) Leaf symptoms of N deficiency in ‘Shiranuhi’. (**B**) Changes in SPAD values of the leaves of ‘Shiranuhi’ according to the period of N deficiency and resupply. (**C**) Changes in leaf N content in ‘Shiranuhi’.

**Figure 2 plants-11-02351-f002:**
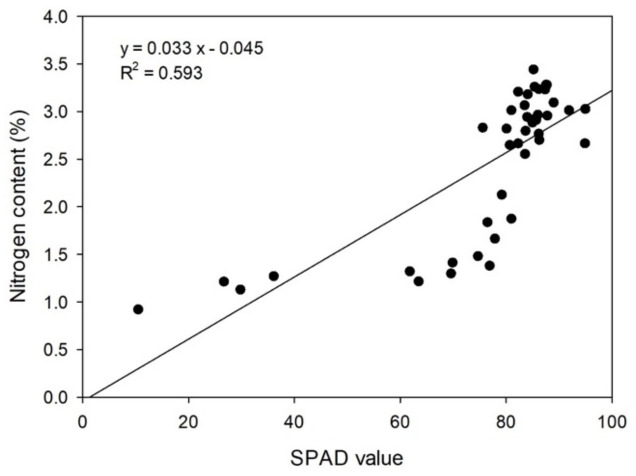
Linear regression of SPAD values and the N content as measured in leaves of ‘Shiranuhi’.

**Figure 3 plants-11-02351-f003:**
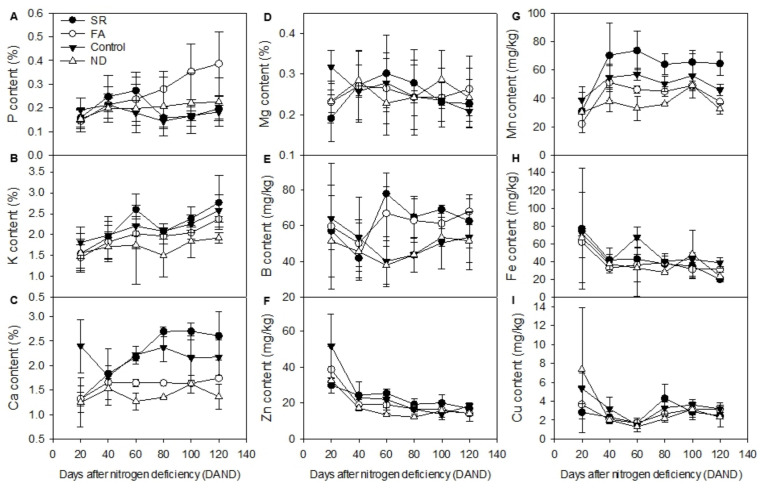
Changes in mineral nutrient concentrations in leaves of ‘Shiranuh’ according to N deficiency and resupply: P (**A**), K (**B**), Ca (**C**), Mg (**D**), B (**E**), Zn (**F**), Mn (**G**), Fe (**H**) and Cu (**I**) concentration.

**Figure 4 plants-11-02351-f004:**
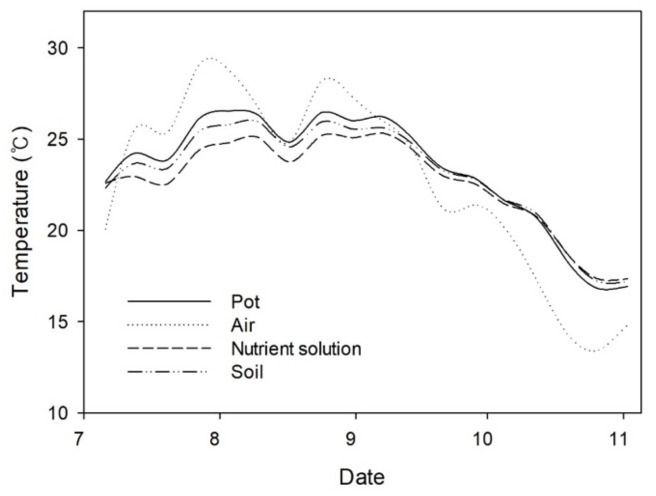
Temperature changes in the pot, nutrient solution, and soil of the hydroponics system and inside the greenhouse during the experimental period.

**Figure 5 plants-11-02351-f005:**
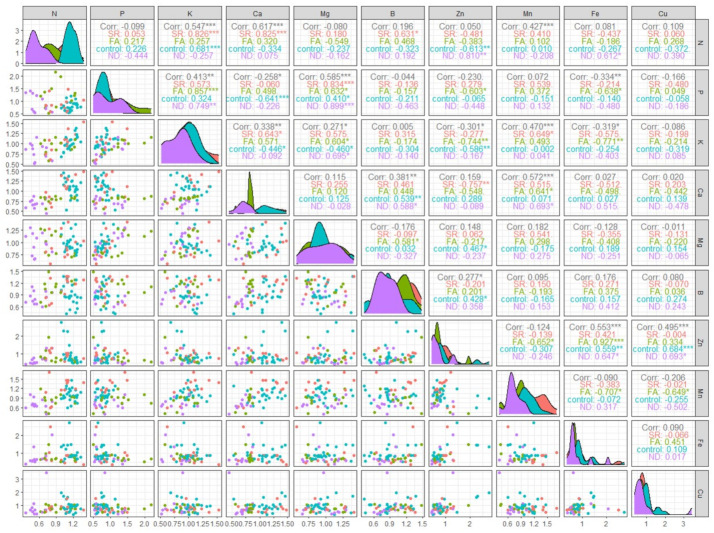
Matrices of Pearson correlation coefficients for the correlations between mineral nutrient contents in the leaves and N contents. In the boxes over the diagonal line, the values of correlation and the significance levels as asterisk were presented. Each significance level is associated with the number of asterisk: *, **, ***; significant at *p* < 0.05, 0.01, 0.001 respectively. The color of the text in the boxes indicates each treatment: orange for SR, green for FA, skyblue for control and purple for ND group.

**Figure 6 plants-11-02351-f006:**
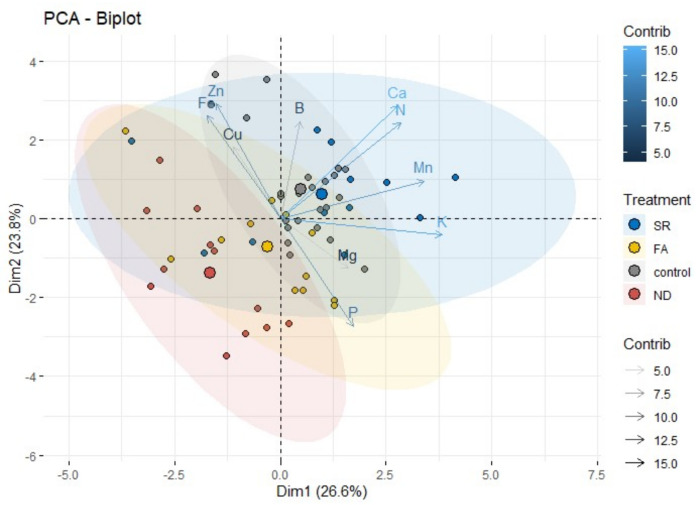
Biplot showing positive and negative correlations of mineral nutrients to the first and second principal components derived from PCA. Dots represent individual plants. Arrows indicate explanatory variables as vectors, and the contribution is represented by their color and shade. The centroid of each ellipse is presented with a larger dot.

**Figure 7 plants-11-02351-f007:**
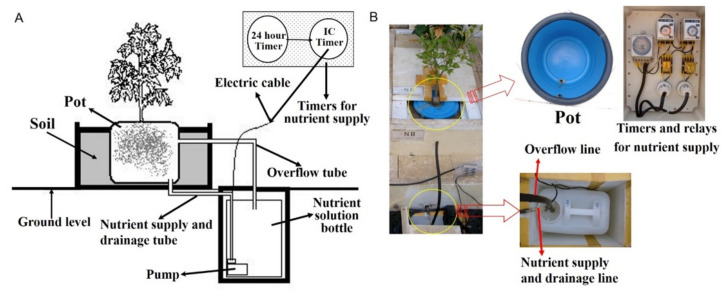
The hydroponics system used for culturing tangor cultivar ‘Shiranuhi’. (**A**) Schematic diagram of the hydroponics system. (**B**) Photos showing the actual set up of the pot, nutrient solution supply equipment, and nutrient solution supply timer.

**Figure 8 plants-11-02351-f008:**
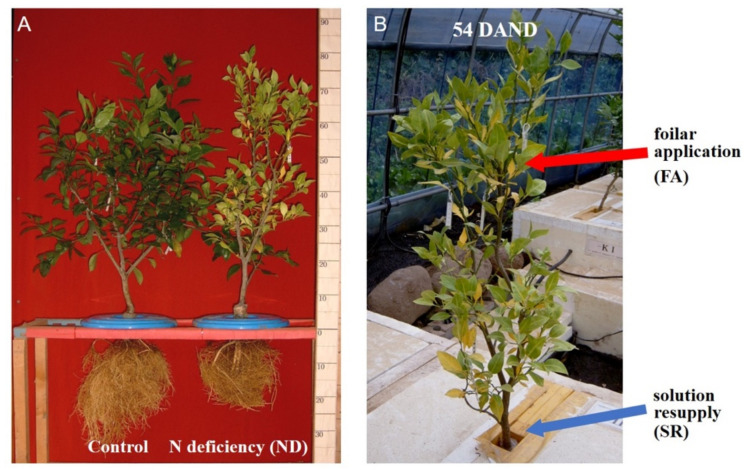
Experimental groups. (**A**) Representative plants in the control and ND groups. (**B**) Representative plants in the FA and SR groups.

**Table 1 plants-11-02351-t001:** Chemical compositions of nutrient solutions used in experiments.

Macronutrients	Concentration (mg L^−1^)	Micronutrients	Concentration (mg L^−1^)
NH_4_-N	18.0 or 0 ^1^	Fe	1.08
NO_3_-N	91.0 or 0	B	0.26
PO_4_-P	20.8	Mn	0.23
K	89.2	Zn	0.25
Ca	72.3	Cu	0.025
Mg	17.8	Mo	0.006
SO_4_-S	23.6		

^1^ The first value is the concentration of the standard solution. The second value is the concentration in the ND group.

**Table 2 plants-11-02351-t002:** Nutrient solution concentration for standard or N deficiency treatment group.

Nutrient Source	Concentration (mg L^−1^)	Nutrient Solution	
		Standard	N Deficiency
KNO_3_	162.8	+	−
5(Ca(NO_3_)_2_·H_2_O)·NH_4_NO_3_	390	+	−
MgSO_4_·7H_2_O	180	+	+
KH_2_PO_4_	91.2	+	+
NH_4_NO_3_	73.6	+	−
CaCl_2_·2H_2_O	264.6	−	+
KCl	120	−	+
Fe-EDTA	8	+	+
H_3_BO_3_	1.5	+	+
MnSO_4_·H_2_O	0.7	+	+
ZnSO_4_·7H_2_O	0.88	+	+
CuSO_4_·5H_2_O	0.1	+	+
Na_2_MoO_4_·2H_2_O	0.015	+	+

## Data Availability

The datasets are available from the corresponding author upon reasonable request.

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
