# Peer review of "Effects of Nitrogen Deficiency and Resupply on the Absorption of Mineral Nutrients by Tangor Cultivar ‘Shiranuhi’ (Citrus unshiu × C. sinensis) Grown in a Hydroponic System"

_plants, 2022, doi:10.3390/plants11182351_

Round 1

Reviewer 1 Report

In this MS, the authors conducted the absorption of various nutrients under N deficiency and N resupply after deficiency. The article is well written, and the findings of the present study will extend our understanding of the cropping technology and physiology of tangor. I request the authors to pay attention to some questions below:

1. There is scope for improvement. Authors need to highlight what is the future of their research and its implications globally.

2. Have the authors considered the effect of the pH of the culture medium?

3. Figure 3 does not label A, B, C, which are mentioned many times in the text (lines 216, 204, and 188).

4. ‘PCA’ in the tile of section 2.5 should be written as ‘Principal Component Analysis’.

5. Years of references are not bolded (lines 459, 464, 482).

Author Response

Reviewer #1

Comments and Suggestions for Authors

In this MS, the authors conducted the absorption of various nutrients under N deficiency and N resupply after deficiency. The article is well written, and the findings of the present study will extend our understanding of the cropping technology and physiology of tangor. I request the authors to pay attention to some questions below:

Comment 1:

There is scope for improvement. Authors need to highlight what is the future of their research and its implications globally.

Thank you for your time to improve this manuscript.

Authors considered the future of tangor hydroponics cultivation as follows and added this sentences to Conclusion section.

“Citrus is the most suitable crop for hydroponics cultivation among fruit crops. Especially, The higher the quality of tangor, the more expensive it is in South Korea. In addition, the export of high-quality tangor continues to increase to Taiwan, China and other countries. Therefore, tangor cultivation using hydroponics is expected to be firmly established as a citrus cultivation method, and it will be spread not only to South Korea but also to many countries in Northeast Asia.”

Comment 2:

Have the authors considered the effect of the pH of the culture medium?

According to reviewer’s recommendation, the sentences were added at materials and methods 3.2.4 section after line 385.

“The nutrient solution was adjusted to pH about 6.0 with 1N NaOH and HCl. During the cultivation, the pH of nutrient solution was periodically checked and adjusted with 1N NaOH to maintain about 6.0”

Comment 3:

Figure 3 does not label A, B, C, which are mentioned many times in the text (lines 216, 204, and 188).

Figure 3 was revised adding labels according to reviewer’s recommendation.

Comment 4:

‘PCA’ in the tile of section 2.5 should be written as ‘Principal Component Analysis’.

According to reviewer’s recommendation, ‘PCA’ was revised.

Comment 5:

  1. Years of references are not bolded (lines 459, 464, 482).

In the case of book citation, the year is not in bold.

Reviewer 2 Report

Plants (ISSN 2223-7747)

Manuscript ID: plants-1877500

Manuscript Title: "Effects of Nitrogen Deficiency and Resupply on the Absorption of Mineral Nutrients by Tangor Cultivar ‘Shiranuhi’ (Citrus unshiu C. sinensis) Grown in a Hydroponic System"

Reviewer comments:

In this study, the authors studied the effects of nitrogen deficiency and resupply on the absorption of mineral nutrients by Tangor Cultivar ‘Shiranuhi’ (Citrus unshiu C. sinensis) Grown in a Hydroponic System”.

The topic is nice and fits well with the scope of Plants, and the results are interest for the scientific community. After carefully reviewing the present work, I feel that this work is suitable for publication after major revision. Therefore, only I have reviewer comments, which might be helpful in improving this work.

Abstract:

L 10-15: “Sweet orange (Citrus sinensis) and mandarin (Citrus reticulata, Citrus unshiu), which are mainly consumed as fresh fruits, and lemon (Citrus limon) and lime (Citrus aurantifolia), which are mostly used in cooking or juice making, are cultivated worldwide. Tangor (C. unshiu × C. sinensis) cultivars obtained through interspecific hybridization have gained popularity in various countries, including South Korea. However, tangor has a relatively short history of cultivation and thus is less well researched in terms of physiology and cropping technology.These sentences are more suitable in the beginning of Introduction part. Pls add a simple or short sentence or sentences on the importance of this study at the beginning of abstract part.

L 15 and 16: “In addition, as tangor grows more vigorously than mandarin and thus has high nutrient demands and a later harvest time, it is more prone to various physiological disorders. Why did the authors compare “Tangor to Mandarin” in the abstract even though the topic of this study was tangor??

L 19-24: “The results showed that, in terms of nutrient absorption, N acted synergistically with K, Ca, and Mn, but antagonistically with P, Mg, Fe, and Zn. Notably, plants under nitrogen deficiency showed characteristic increases in P and Mg contents in the leaves”. The results is very limited, have a broad meaning, and do not meet the requirement of scientific writing in this concern. The results need further clarification in the abstract section.

-        The abstract needs to be further simplified and highlight innovation. After reading the abstract, readers should have a general understanding of the full text and be clear about motivation and breakthrough. Please rewrite the abstract of the paper, which can be discussed from the following aspects:

  • background (This part has been reflected in the first sentence of the abstract), purpose (What are the main scientific problems to be solved in this study?), *
  • method (What new experimental methods were used in this study?),
  • conclusion (For the main purposes mentioned above, whether the desired results have been obtained through this study, and what are the main conclusions?),
  • and significance (What kind of promotion effect does it have on the current relevant research?).
  • In addition, there should not be too much original comparative data in the abstract. People care about the new mechanisms or extraordinary performance more.

-        Numerical data or ratios should be put in place to show the effects of some treatments that have given outstanding results.

-        The abbreviation should be used after the full term. Please be consistent with usage of all abbreviations. Pls revise the abbreviations in the whole parts of MS.

Keywords: I suggest rephrasing some words because keywords should not repeat words from the title.

-      First letter of key words should be capital.

Introduction:

-        The introduction part is long from my opinion. So, pls shorten and avoid repetition.

-        The introduction section require more revision; Pls revise the introduction by consulting the recent relevant references. The manuscript could be substantially improved by citing latest refs.

-        L 48-52: “The effects of N deficiency in citrus have been mainly investigated in sweet orange and mandarin. N deficiency in citrus causes leaves to turn pale green or yellowish and eventually, shed. The leaf veins turn lighter in color and tree growth and development are restricted. The fruits harden, with lower acidity and poorer storability. Finally, fruit yield and quality decrease” Please move this paragraph after the first paragraph of the introduction part.

-        Please highlight more specifically the objective of the work more in the end of introduction section.

Results and discussion

-        DAND; the term have to write in a complete form at the first time and then write the abbreviation only after that.  

-        The results section is too long and contains a link too detailed in all the measurements taken, Please shorten the results section and abbreviate it so as not to prejudice the meaning while avoiding repetition.

-        Resolution of graphical abstract is poor. Kindly improve.

-        The discussion part is almost non-existent, and even in its presence, it is completely insufficient and does not fit with the many results mentioned

-         Some figures are not clear and the quality of the figs is poor and needs a lot of clarification.

-        In the discussion section, conjunctions should be used to show the relationship between sentences.

-        The results present are clearly presented; my main concern is about the discussion, which is not of good level.

-        Please, make an effort to synthesize the text avoiding redundancies and repetitions in the discussion.

-        The discussion sentences need clarification and interpretation, and recent references need to be used as much as possible.

-        This part should be better organized and extended. It is important to try to better deepen and explain.

Material and methods

-        The materials and methods section is well written but relevant references have to add to support them.

-        In the material part, authors must include the greenhouse conditions.

-        The authors have to include the number of plants for each treatment.

-        It is important to add relevant and recent references regarding all methods used in this section.

Conclusions; Do not repeat the above sentences in the conclusion part. In conclusion, you should write a summary of your work in short sentences so that I, as a reader of this article, can understand what the article ended up being.

References;

-        The number of references about 25 ref., only four in the last four years. I think it is good. But, pls delete the old refs., and add the recent ones (2020-2022) and avoid repetition and self-citation as soon as possible.

-        There are several recent ref. in the same trend of the topic of this MS, pls pay attention to this point.

-        Cross check, all the references for mistakes and follow the journal style of reference input.

-        Thoroughly check the typos, syntax errors, and appropriate uses of full forms and their abbreviation.

Comments (minor)

-        The manuscript contains some typo errors; please revise it very carefully. A carful revision for the English Grammar is required. So, language need to be improved thoroughly 

-        Pls, shorten the long sentences and add Ref. as soon as possible with avoiding the repetition.

-        Kindly provide few more recommendation at the end in the form of bullet points.

-        The plagiarism should be reduced according to the journal's requirements.

Author Response

Reviewer #2

Comments and Suggestions for Authors

Reviewer comments:

In this study, the authors studied the effects of nitrogen deficiency and resupply on the absorption of mineral nutrients by Tangor Cultivar ‘Shiranuhi’ (Citrus unshiu C. sinensis) Grown in a Hydroponic System”.

The topic is nice and fits well with the scope of Plants, and the results are interest for the scientific community. After carefully reviewing the present work, I feel that this work is suitable for publication after major revision. Therefore, only I have reviewer comments, which might be helpful in improving this work.

Abstract:

L 10-15: “Sweet orange (Citrus sinensis) and mandarin (Citrus reticulataCitrus unshiu), which are mainly consumed as fresh fruits, and lemon (Citrus limon) and lime (Citrus aurantifolia), which are mostly used in cooking or juice making, are cultivated worldwide. Tangor (C. unshiu × C. sinensis) cultivars obtained through interspecific hybridization have gained popularity in various countries, including South Korea. However, tangor has a relatively short history of cultivation and thus is less well researched in terms of physiology and cropping technology.” These sentences are more suitable in the beginning of Introduction part. Pls add a simple or short sentence or sentences on the importance of this study at the beginning of abstract part.

As reviewer’s recommendation, the abstract section was revised: some sentences (line 10 -12, line 19-22) were deleted and new sentence was added at line 19.

“Based on this result, the cultivation of tangor using hydroponics system will be widely expanded in Northeast Asia and the income of growers will increase through the production of high-quality fruits.”

L 15 and 16: “In addition, as tangor grows more vigorously than mandarin and thus has high nutrient demands and a later harvest time, it is more prone to various physiological disorders. Why did the authors compare “Tangor to Mandarin” in the abstract even though the topic of this study was tangor??

As you know, there is no study on nutrient absorption of tangor, so it is inevitably compared to mandarin reported in the existing literature.

L 19-24: “The results showed that, in terms of nutrient absorption, N acted synergistically with K, Ca, and Mn, but antagonistically with P, Mg, Fe, and Zn. Notably, plants under nitrogen deficiency showed characteristic increases in P and Mg contents in the leaves”. The results is very limited, have a broad meaning, and do not meet the requirement of scientific writing in this concern. The results need further clarification in the abstract section.

As reviewer’s recommendation, the sentences were removed.

- The abstract needs to be further simplified and highlight innovation. After reading the abstract, readers should have a general understanding of the full text and be clear about motivation and breakthrough. Please rewrite the abstract of the paper, which can be discussed from the following aspects:

  • background (This part has been reflected in the first sentence of the abstract), purpose (What are the main scientific problems to be solved in this study?), *
  • method (What new experimental methods were used in this study?),
  • conclusion (For the main purposes mentioned above, whether the desired results have been obtained through this study, and what are the main conclusions?),
  • and significance (What kind of promotion effect does it have on the current relevant research?).
  • In addition, there should not be too much original comparative data in the abstract. People care about the new mechanisms or extraordinary performance more.

- Numerical data or ratios should be put in place to show the effects of some treatments that have given outstanding results.

Since there are several treatments in figures, it is likely that the indicating numerical data makes the figures messy, so it has been removed.

- The abbreviation should be used after the full term. Please be consistent with usage of all abbreviations. Pls revise the abbreviations in the whole parts of MS.

As reviewer’s comments, all abbreviation were checked and revised.

Keywords: I suggest rephrasing some words because keywords should not repeat words from the title.

- First letter of key words should be capital.

As reviewer’s recommendation, Keywords section was revised.

Introduction:

- The introduction part is long from my opinion. So, pls shorten and avoid repetition.

- The introduction section require more revision; Pls revise the introduction by consulting the recent relevant references. The manuscript could be substantially improved by citing latest refs.

- L 48-52: “The effects of N deficiency in citrus have been mainly investigated in sweet orange and mandarin. N deficiency in citrus causes leaves to turn pale green or yellowish and eventually, shed. The leaf veins turn lighter in color and tree growth and development are restricted. The fruits harden, with lower acidity and poorer storability. Finally, fruit yield and quality decrease” Please move this paragraph after the first paragraph of the introduction part.

- Please highlight more specifically the objective of the work more in the end of introduction section.

These sentences (line 48-52) were moved after first paragraph in Introduction section. As reviewer’s recommendation, Introduction part has been reduced.

Results and discussion

- DAND; the term have to write in a complete form at the first time and then write the abbreviation only after that.  

The first sentence of “DAND” was indicated with a complete word at line 94.

- The results section is too long and contains a link too detailed in all the measurements taken, Please shorten the results section and abbreviate it so as not to prejudice the meaning while avoiding repetition.

The result section is modified as short as possible without compromising the meanings.

- Resolution of graphical abstract is poor. Kindly improve.

- The discussion part is almost non-existent, and even in its presence, it is completely insufficient and does not fit with the many results mentioned

- Some figures are not clear and the quality of the figs is poor and needs a lot of clarification.

- In the discussion section, conjunctions should be used to show the relationship between sentences.

- The results present are clearly presented; my main concern is about the discussion, which is not of good level.

- Please, make an effort to synthesize the text avoiding redundancies and repetitions in the discussion.

- The discussion sentences need clarification and interpretation, and recent references need to be used as much as possible.

- This part should be better organized and extended. It is important to try to better deepen and explain.

Based on your comments, figures has been modified and Discussion content has been added.

Material and methods

- The materials and methods section is well written but relevant references have to add to support them.

- In the material part, authors must include the greenhouse conditions.

- The authors have to include the number of plants for each treatment.

- It is important to add relevant and recent references regarding all methods used in this section.

The number of plants for each treatment has already been written at line 400.

Conclusions; Do not repeat the above sentences in the conclusion part. In conclusion, you should write a summary of your work in short sentences so that I, as a reader of this article, can understand what the article ended up being.

Based on your comments, Conclusion content has been added.

References;

- The number of references about 25 ref., only four in the last four years. I think it is good. But, pls delete the old refs., and add the recent ones (2020-2022) and avoid repetition and self-citation as soon as possible.

- There are several recent ref. in the same trend of the topic of this MS, pls pay attention to this point.

- Cross check, all the references for mistakes and follow the journal style of reference input.

- Thoroughly check the typos, syntax errors, and appropriate uses of full forms and their abbreviation.

Based on your comments, old references were removed and recent references were added, but as you know, there is no research paper on the cultivation and physiology of tangor. All citation errors were revised.

Comments (minor)

- The manuscript contains some typo errors; please revise it very carefully. A carful revision for the English Grammar is required. So, language need to be improved thoroughly 

- Pls, shorten the long sentences and add Ref. as soon as possible with avoiding the repetition.

- Kindly provide few more recommendation at the end in the form of bullet points.

- The plagiarism should be reduced according to the journal's requirements.

Round 2

Reviewer 2 Report

Plants (ISSN 2223-7747)-R2

Manuscript ID: plants-1877500

Manuscript Title: "Effects of Nitrogen Deficiency and Resupply on the Absorption of Mineral Nutrients by Tangor Cultivar ‘Shiranuhi’ (Citrus unshiu C. sinensis) Grown in a Hydroponic System"

Reviewer comments:

After revising the modified version of manuscript, I can report that the authors covered some of my previous comments and ignored the others. Therefore, the authors have to pay attention to the following comments:

 Abstract:

-        The abstract needs to be further simplified and highlight innovation. After reading the abstract, readers should have a general understanding of the full text and be clear about motivation and breakthrough. Please rewrite the abstract of the paper, which can be discussed from the following aspects:

  • background (This part has been reflected in the first sentence of the abstract), purpose (What are the main scientific problems to be solved in this study?), *
  • method (What new experimental methods were used in this study?),
  • conclusion (For the main purposes mentioned above, whether the desired results have been obtained through this study, and what are the main conclusions?),
  • and significance (What kind of promotion effect does it have on the current relevant research?).
  • In addition, there should not be too much original comparative data in the abstract. People care about the new mechanisms or extraordinary performance more.

-        Numerical data or ratios should be put in place to show the effects of some treatments that have given outstanding results.

-        The abbreviation should be used after the full term. Please be consistent with usage of all abbreviations. Pls revise the abbreviations in the whole parts of MS.

Introduction:

-        The introduction section requires more revision; Pls revise the introduction by consulting the recent relevant references. The manuscript could be substantially improved by citing latest refs.

-        Please highlight more specifically the objective of the work more in the end of introduction section.

Results and discussion

-        The results section is too long and contains a link too detailed in all the measurements taken, Please shorten the results section and abbreviate it so as not to prejudice the meaning while avoiding repetition.

-        In the discussion section, conjunctions should be used to show the relationship between sentences.

-        Please, make an effort to synthesize the text avoiding redundancies and repetitions in the discussion.

-        The discussion sentences need clarification and interpretation, and recent references need to be used as much as possible.

Material and methods

-        The authors have to include the number of plants for each treatment.

-        It is important to add relevant and recent references regarding all methods used in this section.

References;

-        Cross check, all the references for mistakes and follow the journal style of reference input.

-        Thoroughly check the typos, syntax errors, and appropriate uses of full forms and their abbreviation.

Comments (minor)

-        The manuscript contains some typo errors; please revise it very carefully. A carful revision for the English Grammar is required. So, language need to be improved thoroughly 
